# Arbuscular Mycorrhizal Symbiosis Enhances Photosynthesis in the Medicinal Herb *Salvia fruticosa* by Improving Photosystem II Photochemistry

**DOI:** 10.3390/plants9080962

**Published:** 2020-07-30

**Authors:** Michael Moustakas, Gülriz Bayçu, Ilektra Sperdouli, Hilal Eroğlu, Eleftherios P. Eleftheriou

**Affiliations:** 1Department of Biology, Faculty of Science, Istanbul University, 34134 Istanbul, Turkey; gulrizb@istanbul.edu.tr (G.B.); hilarella87@gmail.com (H.E.); 2Department of Botany, Aristotle University of Thessaloniki, GR-54124 Thessaloniki, Greece; 3Institute of Plant Breeding and Genetic Resources, Hellenic Agricultural Organization-Demeter, Thermi, 57001 Thessaloniki, Greece; ilektras@bio.auth.gr; 4Biology Division, Institute of Graduate Studies in Science, Istanbul University, 34134 Istanbul, Turkey

**Keywords:** sage, inoculation, electron transport rate, *Rhizophagus irregularis*, photoprotective mechanism, redox state, photosynthetic heterogeneity, chlorophyll fluorescence imaging, non-photochemical quenching, medicinal plants

## Abstract

We investigated the influence of *Salvia fruticosa* colonization by the arbuscular mycorrhizal fungi (AMF) *Rhizophagus irregularis* on photosynthetic function by using chlorophyll fluorescence imaging analysis to evaluate the light energy use in photosystem II (PSII) of inoculated and non-inoculated plants. We observed that inoculated plants used significantly higher absorbed energy in photochemistry (Φ*_PSII_*) than non-inoculated and exhibited significant lower excess excitation energy (EXC). However, the increased Φ*_PSII_* in inoculated plants did not result in a reduced non-regulated energy loss in PSII (Φ*_NO_*), suggesting the same singlet oxygen (^1^O_2_) formation between inoculated and non-inoculated plants. The increased Φ*_PSII_* in inoculated plants was due to an increased efficiency of open PSII centers to utilize the absorbed light (F*v*’/F*m*’) due to a decreased non-photochemical quenching (NPQ) since there was no difference in the fraction of open reaction centers (q*_p_*). The decreased NPQ in inoculated plants resulted in an increased electron-transport rate (ETR) compared to non-inoculated. Yet, inoculated plants exhibited a higher efficiency of the water-splitting complex on the donor side of PSII as revealed by the increased F*v*/F*o* ratio. A spatial heterogeneity between the leaf tip and the leaf base for the parameters Φ*_PSII_* and Φ*_NPQ_* was observed in both inoculated and non-inoculated plants, reflecting different developmental zones. Overall, our findings suggest that the increased ETR of inoculated *S. fruticosa* contributes to increased photosynthetic performance, providing growth advantages to inoculated plants by increasing their aboveground biomass, mainly by increasing leaf biomass.

## 1. Introduction

Arbuscular mycorrhizal fungi (AMF) are ubiquitous soil microorganisms that establish mutualistic symbioses with the majority of land plants [1], including most agricultural crops [2]. AMF are considered an important tool in the modern environmentally friendly agriculture in the 21st century for the improvement of crop yield and quality and for the decrease of mineral fertilizers and pesticides/herbicides [3]. The main benefits of AMF to plants include improved acquisition and accumulation of nutrients (such as P and N); and to repay, host plants provide organic carbon to AMF [4]. Photosynthesis-derived CO_2_ assimilation is the main method of production of organic carbon [5] and chlorophyll fluorescence analysis is usually used to characterize photosynthetic performance [6].

Naturally appearing soil microbes may be used as inoculants to maintain crop yields despite reduced resource (water and nutrient) inputs [7]. Several studies revealed that AMF could increase plant biomass and help host plants to improve their nutrient uptake and their tolerance/resistance to biotic and abiotic stresses [5,6,7,8,9,10,11]. Under drought stress AMF inoculation increased the content of compatible solutes, assisting in maintaining the relative water content, and upregulated the antioxidant system of maize plants, facilitating alleviation of oxidative effects through elimination of reactive oxygen species (ROS) [12]. Yet, under drought stress AMF promoted growth, nutrient content, and physiological and biochemical parameters in *Ceratonia siliqua* plants mediating drought tolerance [13].

AMF symbiosis is considered to be the most widespread plant–fungus interaction as it concerns about 90% of terrestrial plant species [1]. Plant inoculation with AMF improves root development and photosynthetic rates, increases nutrient and water uptake, and promotes defense against pathogens [14,15]. The most frequently used species and most studied among members of the *Glomeromycota*, where AMF belong [16,17], is *Rhizophagus irregularis* (Błaszk., Wubet, Renker & Buscot), C. Walker and A. Schüßler (syn. *Glomus irregulare* Błaszk., Wubet, Renker & Buscot, previously known as *Glomus intraradices*) [18,19]. *R. irregularis* is an endomychorrhizal fungus being one of the most popular since it stimulates the growth and development of different plant species, colonizing nearly all the important commercial crops [13,20,21,22]. Recently, an increasing research interest has been noticed in the utilization of AMF for improving plant growth of aromatic and medicinal plants [23,24]. Since medicinal plants are used in diverse productions including herbal, agricultural, pharmaceutical and food, as well as cosmetic industries, AMF colonization research on medicinal plants is of particular value [25]. Greek sage (*Salvia fruticose*) is a perennial herb or sub-shrub, native to the eastern Mediterranean, possessing pharmacological activities with great market demand, being used for its beauty, medicinal and gastronomic value, along with its sweet nectar and pollen.

Since plant production is driven by photosynthesis, evaluating photosynthetic function is a reasonable way to estimate the fate of plant growth and development [26,27], while variations in the efficiency or capacity of photosynthesis can lead to variation in growth rate, productivity and crop yield [28]. Photosynthesis is a highly regulated process in which the absorbed solar energy as photons by the light-harvesting complexes (LHCs) is transferred to the reaction centers (RCs) where through charge separation the electrons flow from photosystem II (PSII) to photosystem I (PSI) [26,27,29] (for details see Figure 1). The two photosystems work coordinately, and the result is the formation of ATP and reducing power (reduced ferredoxin and NADPH) that need to be coordinated with the activity of metabolic processes for carbohydrate synthesis [26,29]. The disturbance of photosynthesis at the molecular level is associated with low electron transport through PSII (ETR) and/or with structural injury to PSII and the LHCs [30]. By using chlorophyll fluorescence imaging analysis, it is possible to measure the fraction of open or closed PSII reaction centers (q*_p_*) and estimate the excess excitation energy (EXC) or, in other words, estimate the effective quantum yield of PSII photochemistry (Φ*_PSII_*) and thus photosynthetic efficiency [30]. Chlorophyll fluorescence analysis that estimates the photosynthetic performance has been frequently used as a highly sensitive indicator of photosynthetic efficiency and an extremely sensitive biomarker to monitor plant health status [31,32,33,34,35,36,37,38]. But since photosynthetic function is not uniform at the whole leaf level, especially under environmental stress conditions [39,40], point chlorophyll fluorescence measurements are not typical of the physiological status of the entire leaf [41,42]. This disadvantage is solved by chlorophyll fluorescence imaging analysis that can reveal the photosynthetic heterogeneity of the entire leaf zone [43,44,45,46,47].

Most researchers who have explored the mechanisms underlying the growth advantage of AMF plants have done it under stress conditions and not under normal growth ones [5,6]. Photosynthetic performance of plant symbiotic relationships with mycorrhizal fungi has been evaluated almost exclusively under stress conditions since soil microbes are used as inoculants to maintain crop yields under decreased water and nutrient inputs [7,13,21,22]. However, the mechanisms that contribute to enhanced photosynthetic performance and plant growth of plant symbiotic relationships with AMF can be better evaluated under non-stress conditions.

In our experiments we wanted to test whether colonization of the medicinal herb *Salvia fruticosa* by *R. irregularis* could result in positive effects on photosynthetic performance and growth of sage plants. We hypothesize that if AMF colonization is successful then plant growth traits will be improved through increased photosynthetic performance. Thus, we applied the method of chlorophyll fluorescence imaging analysis to assess the allocation of absorbed light energy in order to reveal any differentiation mechanism in light energy use that contributes to increased photosynthetic performance of inoculated plants.

## 2. Results

### 2.1. Mycorrhizal Colonization and Plant Growth Performance

Mycorrhizal colonization was estimated by using the gridline intersection method in stained roots from 12 inoculated *S. fruticosa* plants that were cut into about 1 cm long pieces. The percentage of root length colonized was quantified in 15 root segments collected at random per plant. About 69.5 ± 2.2% of the root length of *S. fruticosa* inoculated with *R. irregularis* was colonized, with the formation of vesicles, arbuscules and hyphae. AMF inoculation positively (*p* < 0.05) influenced plant growth traits such as leaf length (Figure 2b), leaf biomass (Figure 3b), and aboveground, biomass (Figure 4a), while the ratio of belowground to aboveground biomass was negatively influenced (Figure 4c) since it decreased (*p* < 0.05) compared to non-inoculated plants. There were no significant changes in the number of leaves (Figure 2a), root length (Figure 2c), stem length (Figure 3a), shoot biomass (Figure 3c), and root biomass (Figure 4b).

### 2.2. The Allocation of Absorbed Light Energy in Inoculated and Non-inoculated Salvia

We estimated the fraction of the absorbed light energy that is used for photochemistry (Φ*_PSII_*), is lost by regulated heat dissipation (Φ*_NPQ_*) and non-regulated energy loss (Φ*_NO_*) [48]. These three quantum yields Φ*_PSII_*, Φ*_NPQ_* and Φ*_NO_*, add up to unity [48]. Φ*_PSII_* in the inoculated *S. fruticosa* increased by 33% (*p* < 0.05) compared to non-inoculated, but the increased Φ*_NPQ_* (38%, *p* < 0.05) in the non-inoculated *Salvia* resulted in non-significant changes in Φ*_NO_* between them (Figure 5).

### 2.3. The Efficiency of PSII in Inoculated and Non-Inoculated Salvia

The maximum efficiency of PSII photochemistry (F*v*/F*m*) did not differ between inoculated and non-inoculated *S. fruticosa* (Figure 6a), but the efficiency of open PSII reaction centers (F*v*’/F*m*’) was 19% (*p* < 0.05) higher in the inoculated ones (Figure 6b), and it was also higher (11%, *p* < 0.05) in the efficiency of the water-splitting complex on the donor side of PSII (F*v*/F*o*) (Figure 6c).

The fraction of open PSII reaction centers, which is the redox state of the plastoquinone pool (q*_p_*), did not differ between inoculated and non-inoculated *S. fruticosa* (Figure 7a). The non-photochemical quenching (NPQ) that reflects heat dissipation of excitation energy was higher (49%, *p* < 0.05) in control (non-inoculated) *S. fruticosa* (Figure 7b). This higher NPQ resulted in significant (*p* < 0.05) lower (33%) electron transport rate (ETR) in the non-inoculated plants (Figure 7c). Overall inoculated *S. fruticosa* presented lower (23%, *p* < 0.05) excess excitation energy (EXC) than non-inoculated (Figure 7d).

### 2.4. Chlorophyll a Fluorescence Images

Representative chlorophyll *a* fluorescence images obtained by the chlorophyll fluorescence imaging analysis method that can reveal any spatial heterogeneity over the leaf are shown in Figure 8 and Figure 9. No significant heterogeneity was detected in the leaf area for the parameters of the minimum chlorophyll *a* fluorescence (F*o*), the maximum chlorophyll *a* fluorescence (F*m*), and the maximum efficiency of PSII photochemistry (F*v*/F*m*) in both inoculated and non-inoculated sage plants (Figure 8). Still, we were able to distinguish a small spatial heterogeneity between the leaf tip and the leaf base of sage for the parameters Φ*_PSII_* and Φ*_NPQ_* (Figure 9).

The average F*o* value of the whole leaf was lower (10%, *p* < 0.05) in inoculated *S. fruticosa* (Figure 8), while average F*m* value did not differ significantly between inoculated and non-inoculated *S. fruticosa,* as well as the average F*v*/F*m* ratio values (Figure 8). Φ*_PSII_* values were higher in the leaf tip of both inoculated and non-inoculated sage, while Φ*_NPQ_* values show the reverse pattern (Figure 9). No spatial heterogeneity over the leaf was observed for Φ*_NO_* images (Figure 9).

## 3. Discussion

AMF inoculation significantly influenced plant growth traits such as aboveground biomass (Figure 4a) mainly by increasing leaf biomass (Figure 3b). However, the ratio of belowground to aboveground biomass decreased (Figure 4c), probably due to the fact that AMF symbiosis improves mineral nutrition and water uptake [6,7,10], thus there is no need for increased root biomass. The increased photosynthetic surface of the host plant provided AMF with organic carbon while the fungi provided improved acquisition and accumulation of nutrients [4] without the need for increased root biomass of the host plant. Similar results were observed with plant growth promoting rhizobacteria [49]. However, AMF inoculation has previously been shown to promote root growth and branching in different plants [50].

In our experiments we used chlorophyll fluorescence imaging analysis in order to reveal the photosynthetic heterogeneity of the entire leaf zone in both inoculated and non-inoculated plants. The observed spatial heterogeneity between the leaf tip and the leaf base of sage for the parameters Φ*_PSII_* and Φ*_NPQ_* (Figure 9) may represent the different developmental zones between the leaf tip and the leaf base [42] and a non-uniform gene expression pattern at the base toward the tip [51]. Heterogeneity in photosynthetic performance depending on the leaf age has been repeatedly described in plant species [52,53,54,55,56,57].

Our data show that the proportion of absorbed energy being used in photochemistry (Φ*_PSII_*) in control (non-inoculated) *S. fruticosa* was lower by 33% compared to inoculated plants (Figure 5). However, this decrease in Φ*_PSII_* was compensated by increases in the photoprotective energy dissipation (Φ*_NPQ_*) that resulted in no difference in Φ*_NO_* between inoculated and non-inoculated plants (Figure 5). Φ*_NO_* comprises of chlorophyll fluorescence internal conversions and intersystem crossing that results to singlet oxygen (^1^O_2_) creation via the triplet state of chlorophyll (^3^chl^*^) [58,59,60,61] (see Figure 1). ^1^O_2_ is considered a highly damaging ROS produced by PSII [62,63,64,65,66] and high levels of ^1^O_2_ activate programmed cell death [67]. However, there was no difference in ^1^O_2_ formation between inoculated and non-inoculated sage plants.

According to the model of PSII function proposed by Genty et al. [68], the increased Φ*_PSII_* in inoculated *S. fruticosa* (Figure 5) can be attributed either to the fraction of open PSII reaction centers (q*_p_*) or to the efficiency of these centers (F*v*’/F*m*’). The former (q*_p_*) is a measure of the redox state of the plastoquinone pool and the latter (F*v*’/F*m*’) is a measure of the supply of energy reaching the PSII reaction centers. In our experiment, considering both the q*_p_* and F*v*’/F*m*’ parameters, it can be proposed that the increased Φ*_PSII_* in inoculated plants was due to a higher efficiency of open PSII centers to utilize the absorbed light (Figure 6b) since the fraction of open reaction centers did not differ between inoculated and non-inoculated sage plants (Figure 7a). Νon-photochemical quenching (NPQ) mechanisms can reduce energy transfer to reaction centers, thus reducing Φ*_PSII_* without any appreciable effect on q*_p_* (so the redox state can be kept relatively steady while a reduced efficiency of PSII centers occur) [69]. The NPQ parameter is an estimate of the dissipated surplus light energy from PSII, primarily representing thermal energy dissipation from LHCII via the zeaxanthin quencher [58,70]. The excess light energy that is dissipated as heat by de-excitation (NPQ) decreases the efficiency of photochemical reactions of photosynthesis (down-regulation of PSII) [56,71,72,73]. Thus, the decreased Φ*_PSII_* in sage plants with non-mycorrhizal inoculum was due to an increased NPQ that reduced the efficiency of PSII centers (F*v*’/F*m*’). An increased NPQ decreases electron-transport rate (ETR), preventing ROS formation [67] (see Figure 1). ROS can contribute directly to PSII damage or inhibit the repair of PSII reaction centers [62,74,75,76,77].

Inoculated *S. fruticosa* plants exhibited significant lower (23%, *p* < 0.05) excess excitation energy (EXC) [78] than non-inoculated (Figure 7d). Efficient photoprotective mechanisms are associated with avoidance of excess energy in chloroplasts [79] as it was observed in the inoculated plants, implying that inoculation contributes to prevention of excess excitation energy at PSII (Figure 7d). It is well defined that NPQ and photoinhibition are strongly interdependent [80,81,82] but how much NPQ or dissipation is needed to successfully limit photoinhibition is a complex question to answer [83,84].

The redox state of the plastoquinone pool did not differ between inoculated and non-inoculated plants, suggesting that it can be kept relatively steady (Figure 7a). Increased available evidence implies that plastoquinone redox state controls stomatal opening in response to light with a more reduced redox state corresponding to increased stomatal opening [85,86,87]. Chloroplast redox regulation has long been considered central for plant photosynthesis through its role in the light-dependent activation of Calvin–Benson–Bassham cycle enzymes [88]. Yet, using chlorophyll fluorescence parameters, carbon assimilation can be easily estimated [89].

Both ratio F*v*/F*m* [90] and its correlated more sensitive form F*v*/F*o* [91] provide an estimate of the potential PSII efficiency of dark-adapted leaves [92]. Though the maximum efficiency of PSII photochemistry (F*v*/F*m*) did not differ between inoculated and non-inoculated plants, the significant difference observed in the efficiency of the water-splitting complex on the donor side of PSII (F*v*/F*o*) suggests that it is a better parameter than F*v*/F*m* and can distinguish small differences in PSII photosynthetic performance [93,94]. In our experiment, the increased F*v*/F*o* ratio in inoculated plants reveals a higher efficiency of the water-splitting complex on the donor side of PSII [95,96]. The donor side photoinhibition mechanism elucidates photoinhibition by malfunction of the water-splitting complex [97,98,99,100]. If the water-splitting complex does not properly reduce the primary electron donor P680^+^, then P680^+^ may cause harmful oxidations in PSII [100].

The maximum chlorophyll *a* fluorescence in the dark-adapted leaf (F*m*) did not differ between inoculated and non-inoculated plants (Figure 8), while the minimum chlorophyll *a* fluorescence in the dark-adapted leaf (F*o*) was lower (10%, *p* < 0.05) in inoculated plants. An increase in F*o* has been considered as a measure of malfunction in the PSII reaction center and decreased efficiency of reaction center photochemistry [70,101,102,103,104,105], and it is observed when the acceptor-side is photoinhibited [106]. Thus, colonization of *S. fruticosa* by the AMF *Rhizophagus irregularis* resulted in a higher efficient PSII donor and acceptor-side.

Overall, our findings suggested that the increased ETR of inoculated *S. fruticosa* (Figure 7c) contributed to increased photosynthetic performance providing growth advantages [107] to inoculated plants by increasing their aboveground biomass (Figure 4a) mainly by increasing leaf biomass (Figure 3b). It seems that the effect of AMF symbiosis on plant growth is coupled with equilibrium between costs and benefits in which the higher carbohydrate cost to plants for AMF symbiosis is balanced by increases in their photosynthetic capacity [108].

## 4. Materials and Methods

### 4.1. Plant Material and Growth Conditions

Seeds of *Salvia fruticosa* Mill. (Greek sage, Lamiaceae) were obtained from Zeytinburnu Medicinal Plant Garden (Istanbul, Turkey). The seeds were surface sterilized in 5% sodium hypochlorite and then washed 3 times with deionized water. The sterilized seeds were then transferred to sterilized filter paper, moistened with deionized water, and left in the dark at room temperature to germinate. The germinated seeds were transferred into plastic multi-pots containing previously sterilized sand and placed in a phytotron with controlled environmental conditions under a long day photoperiod 16 h/8 h, with 70 ± 5%/80 ± 5% humidity (day/night), temperature 23 ± 1 °C/20 ± 1 °C (day/night) and light intensity of 300 ± 20 μmol photons m^−2^ s^−1^ [109]. Salvia seedlings were watered with nutrient solution (Table 1) for 3 weeks and then were transferred to larger pots for arbuscular mycorrhizal application (9 cm base diameter, 15 cm height). All presented data are from two independent biological replicates with three leaf samples (each leaf sample from a different plant) per treatment per experiment for chlorophyll fluorescence measurements and six samples (per treatment per experiment) for growth measurements and for mycorrhizal colonization evaluation.

### 4.2. Arbuscular Mycorrhizal Application

Five g of *Rhizophagus irregularis* (Błaszk., Wubet, Renker & Buscot) C. Walker and A. Schüßler (syn. *Glomus irregulare* Błaszk., Wubet, Renker & Buscot, previously known as *Glomus intraradices*) (Great White Granular 1, 132 propagules per gram, Plant Revolution Inc., Santa Ana, CA, USA) containing 200 spores cm^-3^ were applied to each pot (7 cm depth from the sand surface) in half of the pots before Salvia planting [110]. Treatments included AMF inoculation with *R. irregularis* (inoculated) or the non-mycorrhizal control (control, non-inoculated). Nutrient solution (Table 1) was given (every other day) to both groups for 16 weeks.

### 4.3. Growth Measurements and Mycorrhizal Colonization Evaluation

Roots, stems, and leaves of inoculated or control-non-inoculated *S. fruticosa* plants were harvested separately and washed with running tap water, and tissue lengths and fresh mass were recorded.

The percentage of total root colonization was determined by the gridline intersection method under a bright field microscope [111]. Fresh roots of *S. fruticosa* from 12 plants inoculated with *R. irregularis* were gently washed with running tap water and cut into about 1 cm-long pieces. The specimens were cleared with 10% potassium hydroxide and stained with trypan blue solution according to Phillips and Hayman [112]. Then they were observed under a microscope and the percentage of root length colonized was quantified in 15 root segments collected at random per plant.

### 4.4. Chlorophyll Fluorescence Imaging Analysis

Chlorophyll fluorescence parameters were measured in dark-adapted (20 min) randomly selected Salvia leaves of the same developmental stage (control-non-inoculated or inoculated) using a chlorophyll fluorometer imaging-PAM M-Series (Heinz Walz GmbH, Effeltrich, Germany) as previously described [113]. In each leaf 9 areas of interest (AOI) were selected for analysis. The initial chlorophyll *a* fluorescence (F*o*) was obtained by modulated measuring light of 0.5 μmol photons m^–2^ s^−1^ and the maximal fluorescence (F*m*) with a saturating pulse (SP) of 6000 μmol photons m^–2^ s^−1^ (see Figure 10 for explanation). The maximum efficiency of PSII photochemistry (F*v*/F*m*) was calculated as (F*m*-F*o*)*/*F*m* and the efficiency of the water-splitting complex on the donor side of PSII (F*v*/F*o*) was calculated as (F*m*-F*o*)*/*F*o* [114]. The steady-state photosynthesis was measured after 5 min illumination time before switching off the actinic light (AL) of 200 μmol photons m^–2^ s^−1^ (Figure 10). The maximum chlorophyll *a* fluorescence in the light-adapted leaf (F*m*΄) was measured with SPs every 20 s for 5 min after application of the AL (200 μmol photons m^–2^ s^−1^). The effective quantum yield of PSII photochemistry (Φ*_PSII_*) was calculated by the Imaging Win software (Heinz Walz GmbH, Effeltrich, Germany) as (F*m*’-F*s*)/F*m*’, and the redox state of the plastoquinone pool (q*_p_*), which is the fraction of open PSII reaction centers as (F*m*’-F*s*)/(F*m*’-F*o*’). The minimum chlorophyll *a* fluorescence in the light-adapted leaf (F*o*’) was computed by the Imaging Win software using the approximation of Oxborough and Baker [115] as F*o*’= F*o*/(F*v*/F*m*+F*o*/F*m*’). The quantum yield of regulated non-photochemical energy loss in PSII (Φ*_NPQ_*) was calculated as F*s*/F*m*’−F*s*/F*m* and the quantum yield of non-regulated energy loss in PSII (Φ*_NO_*) as F*s*/F*m*. The efficiency of excitation energy capture by open PSII reaction centers (F*v*’/F*m*’) was calculated as (F*m*’-F*o*’)*/*F*m*’, and non-photochemical quenching (NPQ) that reflects heat dissipation of excitation energy as (F*m*-F*m*’)/F*m*’. The relative PSII electron transport rate (ETR) was calculated as Φ*_PSII_* x PAR x c x abs, by the Imaging Win software, where PAR is the Photosynthetic Active Radiation (200 μmol photons m^–2^ s^−1^), c is 0.5 since the absorbed light energy is assumed to be equally distributed between PSII and PSI, and abs is the total light absorption of the leaf taken as 0.84. The relative excess energy at PSII was calculated according to Bilger et al. [78], as EXC = (F*v*/F*m*−Φ*_PSII_*)/(F*v*/F*m*).

Representative chlorophyll fluorescence color-coded images of the initial chlorophyll *a* fluorescence (F*o*), the maximal fluorescence (F*m*), the maximum efficiency of PSII photochemistry (F*v*/F*m*), the effective quantum yield of PSII photochemistry (Φ*_PSII_*), the quantum yield of regulated non-photochemical energy loss in PSII (Φ*_NPQ_*) and the quantum yield of non-regulated energy loss in PSII (Φ*_NO_*) are also displayed for control-non-inoculated and inoculated *Salvia fruticosa* leaves.

### 4.5. Statistical Analysis

Chlorophyll fluorescence parameters represent averaged values (n = 6) from two independent experiments with three leaf samples (each leaf sample from a different plant) per treatment per experiment, while growth measurements and mycorrhizal colonization evaluation represent averaged values (n = 12) from six samples (per treatment per experiment). Results are expressed as mean ± SD. Statistically significant differences between the treatments were analyzed by the Student’s *t*-test at a level of *p* < 0.05 (*StatView* computer package, Abacus Concepts, Inc Berkley, CA, USA).

## Figures and Tables

**Figure 1 plants-09-00962-f001:**
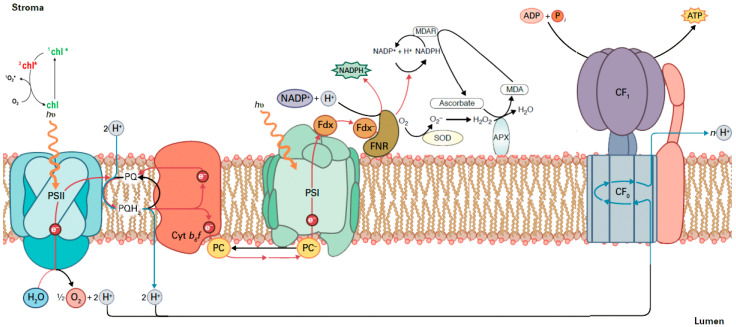
The chloroplast electron transport chain from photosystem II (PSII) to photosystem I (PSI) and finally to ferredoxin (Fdx) to form NADPH, showing also the creation of singlet oxygen (^1^O_2_) via the triplet state of chlorophyll (^3^chl^*^), other reactive oxygen species (ROS) formation and scavenging, ATP synthesis and the oxidation at PSII (water-splitting complex) of water to O_2_, electrons (e^–^), and protons (H^+^). Electrons are transferred from H_2_O to NADP^+^ while accompanying this electron transfer, and a proton gradient is established across the membrane utilized for the synthesis of ATP by the ATP synthase. Light-harvesting complex of PSII (LHCII) absorbs light energy and transfers it to the reaction center of PSII, where excitation of specially bound chlorophyll (Chl) molecules results in transfer of an electron from H_2_O oxidation to quinone A (Q_A_). The fully reduced quinol molecule (PQH2) picks up two protons from the stroma and is oxidized to a quinone (PQ) and while the electrons are transferred through cytochrome b_6_f, to plastocyanin (PC) and to PSI, protons are transferred from the stroma to the chloroplast lumen. Shown are the structures of the soluble proteins ferredoxin (Fdx) and ferredoxin-NADP^+^ reductase (FNR), on the stromal side, that transfer the electrons to NADP^+^ to form NADPH. When NADP^+^ is not available (e.g., NADPH is not used in Calvin–Benson–Bassham cycle), electrons are transferred to molecular oxygen (O_2_) forming superoxide anions (O_2_**^•^****^–^**) that are converted by the superoxide dismutase (SOD) to hydrogen peroxide (H_2_O_2_) that is reduced by ascorbate peroxidase (APX) to water and oxygen. APX uses electrons from ascorbate that are oxidized, but through monodehydroascorbate reductase (MDAR) ascorbate are reduced from NADPH, thus contributing to NADP^+^ availability [Modified from Biochemistry & Molecular Biology of Plants, second edition 2015, Bob B. Buchanan, Wilhelm Gruissem and Russell L. Jones (eds), John Wiley & Sons, Ltd. (after license)].

**Figure 2 plants-09-00962-f002:**
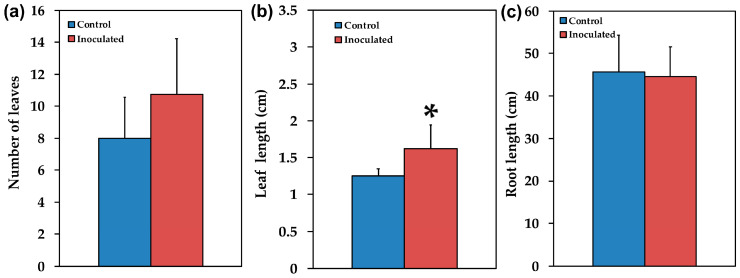
The number of leaves (**a**), the leaf length in cm (**b**), and the root length in cm (**c**), of control (non-inoculated) and inoculated *Salvia fruticosa* plants. Error bars on columns are standard deviations (n = 12). An asterisk (*) represents a significantly different mean between the two treatments by Student’s *t*-test at a level of *p* < 0.05.

**Figure 3 plants-09-00962-f003:**
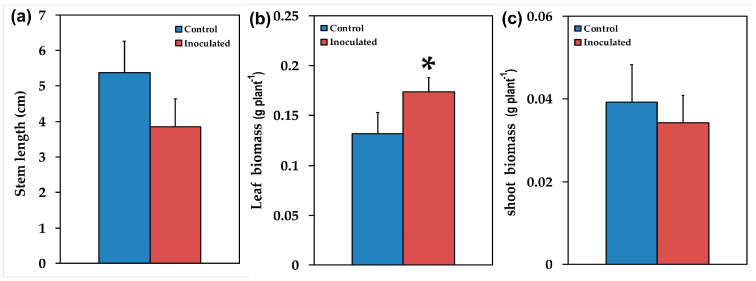
The stem length in cm (**a**), the leaf fresh biomass in g per plant (**b**), and the shoot fresh biomass in g per plant (**c**), of control (non-inoculated) and inoculated *Salvia fruticosa* plants. Error bars on columns are standard deviations (n = 12). An asterisk (*) represents a significantly different mean between the two treatments by Student’s *t*-test at a level of *p* < 0.05.

**Figure 4 plants-09-00962-f004:**
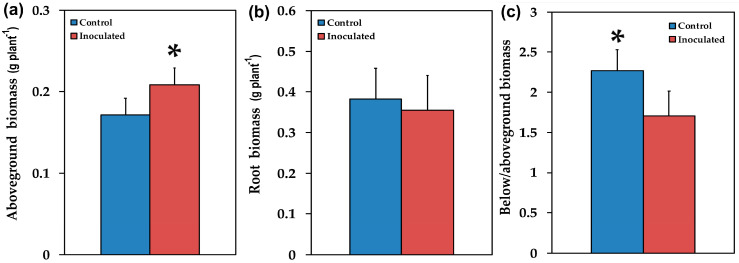
The aboveground fresh biomass in g per plant (**a**), the root fresh biomass in g per plant (**b**), and the ratio of belowground to aboveground fresh biomass (**c**), of control (non-inoculated) and inoculated *Salvia fruticosa* plants. Error bars on columns are standard deviations (n = 12). An asterisk (*) represents a significantly different mean by Student’s *t*-test at a level of *p* < 0.05.

**Figure 5 plants-09-00962-f005:**
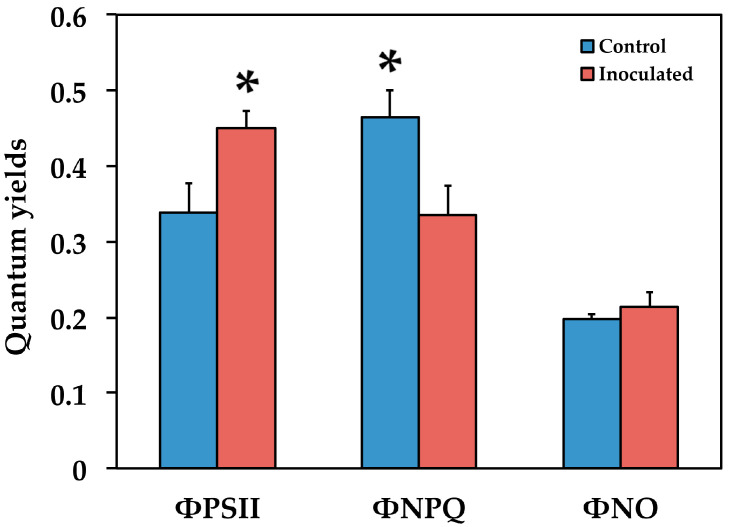
The quantum yields of PSII photochemistry (Φ*_PSII_*) of regulated non-photochemical energy loss in PSII (Φ*_NPQ_*) and of non-regulated energy loss in PSII (Φ*_NO_*) of *Salvia fruticosa* leaves from control (non-inoculated) and inoculated plants. Error bars on columns are standard deviations (n = 6). An asterisk (*) represents a significantly different mean for the same parameter by Student’s *t*-test at a level of *p* < 0.05.

**Figure 6 plants-09-00962-f006:**
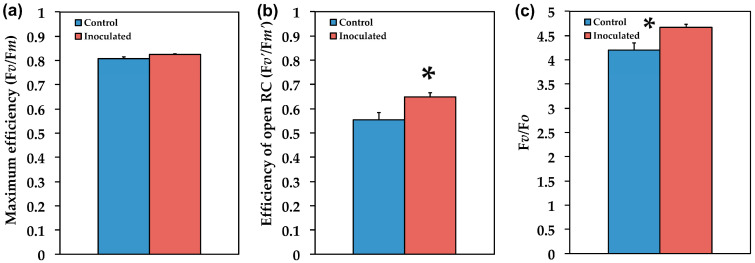
The maximum efficiency of PSII photochemistry (F*v*/F*m*) (**a**), the efficiency of open PSII reaction centers (F*v*’/F*m*’) (**b**), and the efficiency of the water-splitting complex on the donor side of PSII (F*v*/F*o*) (**c**), of Salvia fruticosa leaves from control (non-inoculated) and inoculated plants. Error bars on columns are standard deviations (n = 6). An asterisk (*) represents a significantly different mean between the two treatments by Student’s *t*-test at a level of *p* < 0.05.

**Figure 7 plants-09-00962-f007:**
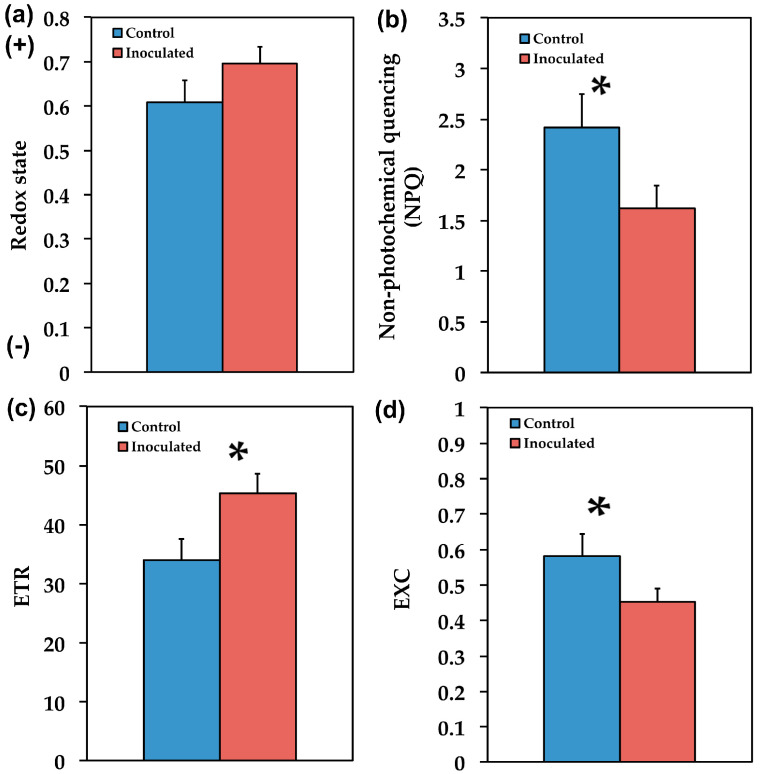
The redox state of plastoquinone pool (q*_p_*) which is a measure of the fraction of open PSII reaction centers (**a**), the non-photochemical quenching (NPQ) that reflects heat dissipation of excitation energy (**b**), the relative PSII electron transport rate (ETR) (**c**)**,** and the excess excitation energy (EXC) (**d**)**,** of *Salvia fruticosa* leaves from control (non-inoculated) and inoculated plants. Error bars on columns are standard deviations (n = 6). An asterisk (*) represents a significantly different mean between the two treatments by Student’s *t*-test at a level of *p* < 0.05.

**Figure 8 plants-09-00962-f008:**
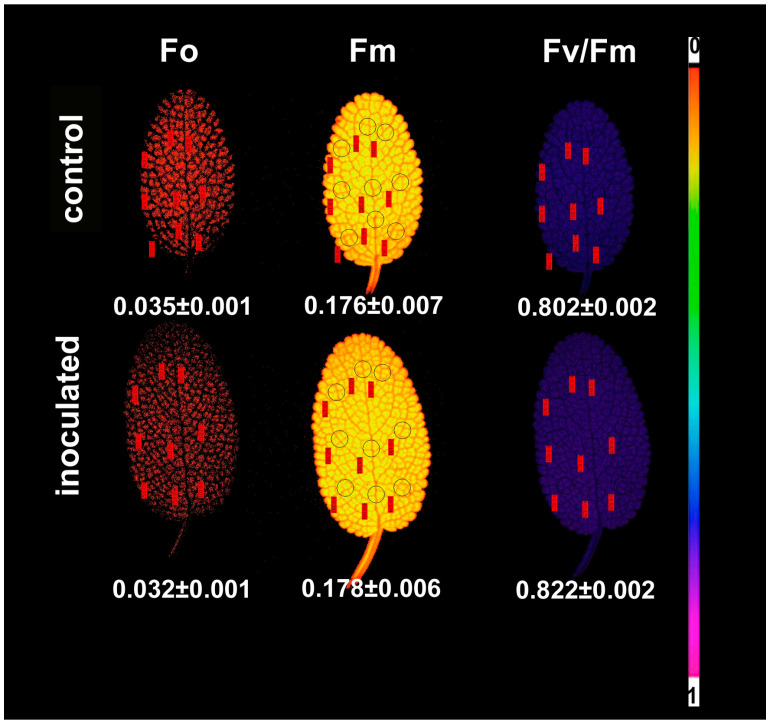
Representative chlorophyll *a* fluorescence images of the minimum chlorophyll *a* fluorescence in the dark-adapted leaf (F*o*), the maximum chlorophyll *a* fluorescence in the dark-adapted leaf (F*m*), and the maximum efficiency of PSII photochemistry (F*v*/F*m*) of *Salvia fruticosa* leaves from control (non-inoculated) and inoculated plants. The color code depicted at the right side of the image ranges from black (pixel values 0.0) to purple (1.0). The nine areas of interest (AOI) are shown in each image. The average value (±SD) of the whole leaf for each parameter is shown.

**Figure 9 plants-09-00962-f009:**
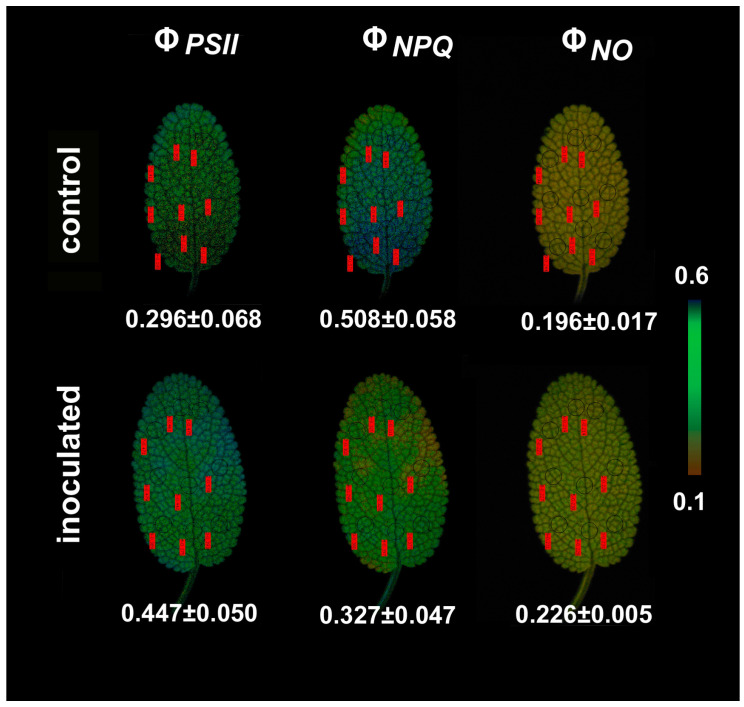
Representative chlorophyll *a* fluorescence images of the effective quantum yield of PSII photochemistry (Φ*_PSΙΙ_*), the quantum yield of regulated non-photochemical energy loss in PSII (Φ*_NPQ_*), and the quantum yield of non-regulated energy loss in PSII (Φ*_NO_*) of *Salvia fruticosa* leaves from control (non-inoculated) and inoculated plants. The color code depicted at the right side of the image ranges from values 0.1 to 0.6. The nine areas of interest (AOI) are shown in each image. The average value (± SD) of the whole leaf for each parameter is shown.

**Figure 10 plants-09-00962-f010:**
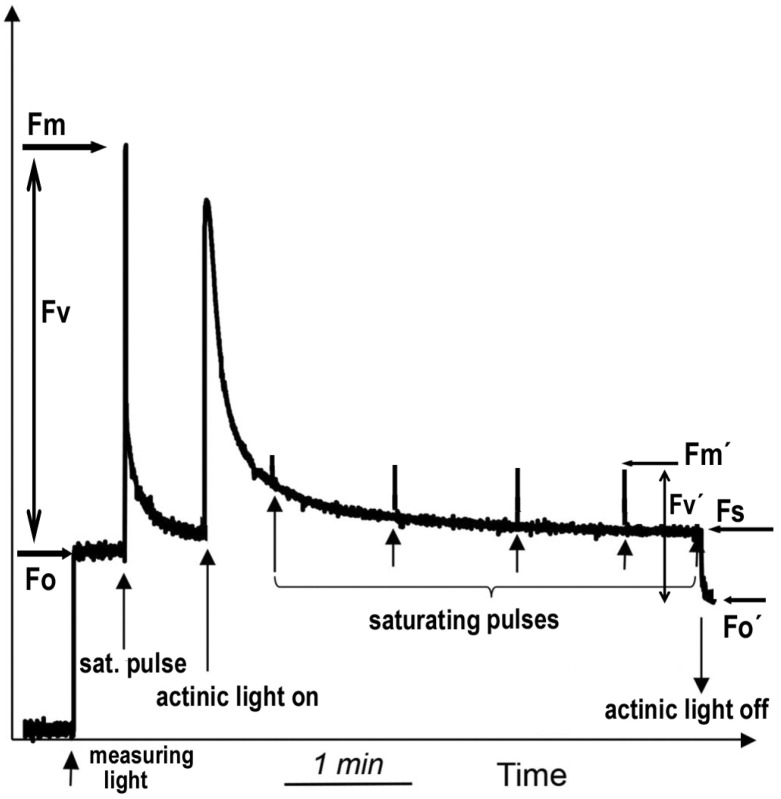
A typical modulated fluorescence trace using dark-adapted leaf material showing how F*o*, F*m*, F*o*’, F*m*’ and F*s* are formed to measure photochemical and non-photochemical parameters. In the dark-adapted state, a “measuring light” is switched on that is of too low intensity to induce electron transport through PSII but high enough to elicit the minimal level of chlorophyll fluorescence, termed F*o*. A brief saturating pulse of light results in the formation of the maximum yield of fluorescence, F*m*. The difference between F*m* and F*o* is the variable fluorescence, F*v*. The ratio F*v*/F*m* is indicator of the maximum quantum yield of PSII photochemistry. The application of saturating pulses under actinic light illumination closes all the reaction centers and provides the maximum fluorescence in the light-adapted state, termed F*m*’. The steady-state level of fluorescence in the light is termed F*s* and is measured immediately before switching off the actinic light. F*o*’ is measured immediately after switching off the actinic light. The difference between F*m*’ and F*o*’ is the variable fluorescence, F*v*’. The ratio F*v*’/F*m*’ is an indicator of the efficiency of excitation energy captured by open PSII reaction centers.

**Table 1 plants-09-00962-t001:** The composition of the nutrient solution used in the experiment (pH 6.5).

Nutrient Elements	Stock Solution (g L^-1^)	Used in Nutrient Solution (ml L^-1^)
KH_2_PO_4_	32.93	2.00	
K_2_SO_4_	29.07	2.00	
MgSO_4_.7H_2_O	30.42	4.00	
CaCl_2_.2H_2_O	11.00	4.00	
NH_4_NO_3_	56.00	5.00	
MnCl_2_.4H_2_O	1.443	1.00	
NaMoO_4_.2H_2_O	0.018	1.00	
H_3_BO_3_	0.018	1.00	
CuSO_4_.5H_2_O	0.008	1.00	
ZnSO_4_.7H_2_O	1.44	1.00	
FeSO_4_.7H_2_O + + Titriplex III EDTA	7.00+9.30	1.00

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
