# Peer review of "Arbuscular Mycorrhizal Symbiosis Enhances Photosynthesis in the Medicinal Herb Salvia fruticosa by Improving Photosystem II Photochemistry"

_plants, 2020, doi:10.3390/plants9080962_

Round 1
Reviewer 1 Report
Your data might have an originality to publish in Plants. However, the results of your experiment were only photosynthetic parameters. In my opinion, the most important data for your experiment was effect of arbuscular mycorrhizal fungi. Unfortunately, there was no data on the colonization of arbuscular mycorrhizal fungi. This data is essential to estimate the effect of inoculation. Also, I predict that nutrients and various ingredients of photosynthetic apparatus (e.g. chlorophyll) are accelerated by inoculation of arbuscular mycorrhizal fungi. Unless you show these data, you cannot explain the cause to increase various photosynthetic parameters. Therefore, I thought that your submission was unacceptable.
Moreover, I would like to indicate some problems of your manuscript as following;
- The information of Salvia fruticose was quite scanty. I could not understand the inevitably to use this species.
- Light intensity was weak for the cultivation of plants. Also, you must show the component of nutrient solution. Quotation of component was invalid.
Author Response
Your data might have an originality to publish in Plants. However, the results of your experiment were only photosynthetic parameters. In my opinion, the most important data for your experiment was effect of arbuscular mycorrhizal fungi. Unfortunately, there was no data on the colonization of arbuscular mycorrhizal fungi. This data is essential to estimate the effect of inoculation. Also, I predict that nutrients and various ingredients of photosynthetic apparatus (e.g. chlorophyll) are accelerated by inoculation of arbuscular mycorrhizal fungi. Unless you show these data, you cannot explain the cause to increase various photosynthetic parameters. Therefore, I thought that your submission was unacceptable.
Response: In our revised manuscript we provided data regarding AMF colonization and plant growth traits. We also correlate and explain the increase in the photosynthetic parameters (mainly ETR) with plant growth data (Lines 187-193, 265-270).
Moreover, I would like to indicate some problems of your manuscript as following;
The information of Salvia fruticose was quite scanty. I could not understand the inevitably to use this species.
Response: In our revised manuscript (Lines 66-72) we provided the reason of choosing Salvia fruticosa.
Light intensity was weak for the cultivation of plants. Also, you must show the component of nutrient solution. Quotation of component was invalid.
Response: In our revised manuscript we corrected the light intensity of plant cultivation that was 300 μmol photons m–2 s−1 as used also in our previous article [109]. Thank you for your comment that helped us to find our mistake. A table was included with the nutrient solution components.
Reviewer 2 Report
In my opinion, each part of the article is written clearly. Authors should correct the journal name in position 7 on italics. I recommend the article for publication in this form.
Author Response
In my opinion, each part of the article is written clearly. Authors should correct the journal name in position 7 on italics. I recommend the article for publication in this form.
Response: We would like to thank you for your comments and the notice to correct the journal name in position 7 on italics
Reviewer 3 Report
What is the main question addressed by the research? -If AMF colonization of Salvia could take place and to evaluate the influence on photosynthetic performance. Is it relevant and interesting? -As stated previously, yes to both. How original is the topic? -Average, in the sense of carbon cost, but interesting method in the use of chlorophyll fluorescence to evaluate. What does it add to the subject area compared with other published material? -It is not a major leap forward, but an interesting piece to a much larger puzzle of how to evaluate the 'cost' of AMF associations. Is the paper well written? -As stated previously, simply no. An English and grammar check would help. The entire document could be shortened considerably and tightened.Is the text clear and easy to read? -No. Are the conclusions consistent with the evidence and arguments presented? Do they address the main question
posed to support the recommendation ##"Minor"##. -Yes, but could be tightened throughout. The paper is interesting but could do with an extensive reread for English and Grammar. The introduction is disjointed and could do with a rewrite and could be shortened and made more direct. I feel this could be could be done to the entire paper. Thank you.
Author Response
What is the main question addressed by the research? -If AMF colonization of Salvia could take place and to evaluate the influence on photosynthetic performance. Is it relevant and interesting? -As stated previously, yes to both. How original is the topic? -Average, in the sense of carbon cost, but interesting method in the use of chlorophyll fluorescence to evaluate. What does it add to the subject area compared with other published material? -It is not a major leap forward, but an interesting piece to a much larger puzzle of how to evaluate the 'cost' of AMF associations. Is the paper well written? -As stated previously, simply no. An English and grammar check would help. The entire document could be shortened considerably and tightened.
Is the text clear and easy to read? -No. Are the conclusions consistent with the evidence and arguments presented? Do they address the main question posed to support the recommendation ##"Minor"##. -Yes, but could be tightened throughout. The paper is interesting but could do with an extensive reread for English and Grammar. The introduction is disjointed and could do with a rewrite and could be shortened and made more direct. I feel this could be could be done to the entire paper. Thank you.
Response: In our revised manuscript we provided data regarding AMF colonization and plant growth traits. We have rewritten Abstract, Introduction, Results, Discussion and Materials and Methods taking into consideration your comments as well as the comments of other reviewers. We corrected the English language throughout the manuscript.
Reviewer 4 Report
The authors have layed out an experiment to determine how VAM affect photosynthesis of sage as compared to controls under non-stress conditions, which authors claim to be rare in literature. The relevance of the study is high as photosynthesis is the entrance of C to the food chain and VAM are key to increase production under adverse conditions in agriculture, to feed the increasing world population. So the relevance of the topic is evident.
However, photosynthesis is a hardly known process and in order to increase impact of this article, some basic physiological processess need to be included in the article. First in the introduction, where no comments to the parametres that are to be discussed is anticipated. Secondly in the materials and methods, although description of parametres is perfect, most people do not know what all those parametres refer to, so an image needs to be included, highlighting FV, Fo and so on, so the parametres can be understood. Then, in the results, an image with PSII, oxygen forming complex, secondary carriers, PSI, LHCII, xanthophylx, oxygen and ROS is necessary to illustrate all complexity that is perfectly discussed but impossible to understand unless you have it fresh in your brain. So, two figures to be included
Data of mycorrhization and growth needs to be included. Essential oil content will really increase soundness of the work and make it more interesting. Otherwise, the article is to be rejected as there is no connection between photosynthesis and plant performance, just a plain measure of chlorophyl fluorescence in two sides of the leaves, confirming that they are different as has been demosntrated before.
Abstract needs rewrited, describing all data refered to mycorrhyzal plants, not changing between controls and mycorrhizal plants. Make positive statements in order to highlight the benefits of the mycorryza
In the introduction, remove references to heavy metals as this information is out of the scope of this work. They are highlighted in the text.

Author Response
The authors have layed out an experiment to determine how VAM affect photosynthesis of sage as compared to controls under non-stress conditions, which authors claim to be rare in literature. The relevance of the study is high as photosynthesis is the entrance of C to the food chain and VAM are key to increase production under adverse conditions in agriculture, to feed the increasing world population. So the relevance of the topic is evident.
Response: We would like to thank you for your general comments as well as for the comments directly on the manuscript that really helped us a lot to improve our manuscript. In the revised manuscript we have included all your suggestions and comments written on the text as well as we addressed all your general comments. We believe that it has been substantially improved due to your comments. We thank you once more for your time and efforts that were very helpful to us.
However, photosynthesis is a hardly known process and in order to increase impact of this article, some basic physiological processess need to be included in the article. First in the introduction, where no comments to the parametres that are to be discussed is anticipated.
Response: We have included in the introduction a paragraph with comments to some of the parametres that will be discussed in the manuscript and extended the text on photosynthesis.
Secondly in the materials and methods, although description of parametres is perfect, most people do not know what all those parametres refer to, so an image needs to be included, highlighting FV, Fo and so on, so the parametres can be understood.Then, in the results, an image with PSII, oxygen forming complex, secondary carriers, PSI, LHCII, xanthophylx, oxygen and ROS is necessary to illustrate all complexity that is perfectly discussed but impossible to understand unless you have it fresh in your brain. So, two figures to be included
Response: We have included in the Materials and Methods a Figure highlighting Fv, Fo and so on and explained the basic methodology in chlorophyll fluorescence measurements. However, we have not included the second Figure you proposed that together with the first Figure will turn out our manuscript from a research paper to rather a review one.
Data of mycorrhization and growth needs to be included. Essential oil content will really increase soundness of the work and make it more interesting. Otherwise, the article is to be rejected as there is no connection between photosynthesis and plant performance, just a plain measure of chlorophyl fluorescence in two sides of the leaves, confirming that they are different as has been demosntrated before.
Response: In our revised manuscript we provided data regarding AMF colonization and plant growth traits. We also correlate and explain the increase in the photosynthetic parameters (mainly ETR) with plant growth data (Lines 187-193, 265-270).
We agree that essential oil content will really increase soundness of the work and make it more interesting but we don’t have such data.
Abstract needs rewrited, describing all data refered to mycorrhyzal plants, not changing between controls and mycorrhizal plants. Make positive statements in order to highlight the benefits of the mycorryza
Response: Yes, you are absolutely right. The Abstract was rewritten according to your comment.
In the introduction, remove references to heavy metals as this information is out of the scope of this work. They are highlighted in the text.
Response: Yes, we removed references to heavy metals from introduction.

Round 2
Reviewer 1 Report
On the revised manuscript, you added the data on the colonization of arbuscular mycorrhizal fungi. Moreover, you added verious data in revised manuscript. Therefore, I admit the novelty of your research. I think that your manuscript is acceptable in progress.
Before acceptance, I would like to confirm whether you have the data of dry mass or not. If possible, it is better to change into the data of dry mass. Also, you should change “weight” into “mass” because weight was affected by gravity.
Author Response
Before acceptance, I would like to confirm whether you have the data of dry mass or not. If possible, it is better to change into the data of dry mass. Also, you should change “weight” into “mass” because weight was affected by gravity.
Response: Unfortunately we have only fresh mass data. “Weight” was changed to “mass”.
Reviewer 4 Report
The authors have addresed most comments in their review. However, some more amendments are to be made.
mycorrhization: how many roots of which length were evaluated? the method is described but the percentage appears from nowhere.
figures. I think the second figure I requested needs to be included and the ms. will not be transformed in a review ms. Rather it will be improved and raise interest for readers.
First paragraph of discussion. It is evident to this reviewer that the increase in shoot biomass caused by the mycorrhiza intends to increase the photosynthetic surface to increase exudate rate to feed the fungi,while the fungi will provide the increased absorption surface. Probably several references are available to support this statement; the one I know supports this effect for bacteria by Gutierrez Mañero et al, 2001 PHYSIOLOGIA PLANTARUM 111: 206–211. 2001
Author Response
mycorrhization: how many roots of which length were evaluated? the method is described but the percentage appears from nowhere.
Response: In the revised manuscript we provided details of colonization also in the Results section.
figures. I think the second figure I requested needs to be included and the ms. will not be transformed in a review ms. Rather it will be improved and raise interest for readers.
Response: We finally adopted your suggestion to include the second figure. It seems that the manuscript was improved. We thank you for your help in improving our manuscript.
First paragraph of discussion. It is evident to this reviewer that the increase in shoot biomass caused by the mycorrhiza intends to increase the photosynthetic surface to increase exudate rate to feed the fungi,while the fungi will provide the increased absorption surface. Probably several references are available to support this statement; the one I know supports this effect for bacteria by Gutierrez Mañero et al, 2001 PHYSIOLOGIA PLANTARUM 111: 206–211. 2001
Response: We improved the first paragraph of discussion and we included this citation.